# Strain-tunable Berry curvature in quasi-two-dimensional chromium telluride

Hang Chi [1,2] ✉, Yunbo Ou [1] ✉, Tim B. Eldred [3], Wenpei Gao [3],
Sohee Kwon[4], Joseph Murray[5], Michael Dreyer [5], Robert E. Butera [6],
Alexandre C. Foucher[7], Haile Ambaye[8], Jong Keum [8,9], Alice T. Greenberg[2],
Yuhang Liu[4], Mahesh R. Neupane[2,4], George J. de Coster[2], Owen A. Vail[2],
Patrick J. Taylor[2], Patrick A. Folkes[2], Charles Rong[2], Gen Yin[10], Roger K. Lake [4],
Frances M. Ross [7], Valeria Lauter [8], Don Heiman[1,11] &
Jagadeesh S. Moodera[1,12] ✉

Magnetic transition metal chalcogenides form an emerging platform for exploring spin-orbit driven Berry phase phenomena owing to the nontrivial interplay between topology and magnetism. Here we show that the anomalous Hall effect in pristine $Cr_2Te_3$ thin films manifests a unique temperature-dependent sign reversal at nonzero magnetization, resulting from the momentum-space Berry curvature as established by first-principles simulations. The sign change is strain tunable, enabled by the sharp and well-defined substrate/film interface in the quasi-two-dimensional $Cr_2Te_3$ epitaxial films, revealed by scanning transmission electron microscopy and depth-sensitive polarized neutron reflectometry. This Berry phase effect further introduces hump-shaped Hall peaks in pristine $Cr_2Te_3$ near the coercive field during the magnetization switching process, owing to the presence of strain-modulated magnetic layers/domains. The versatile interface tunability of Berry curvature in $Cr_2Te_3$ thin films offers new opportunities for topological electronics.

In recent years, a variety of novel two-dimensional (2D) van der Waals magnets have been discovered, founding the active field of 2D magnetism[1]. Among these prospective compounds, binary chromium tellurides $Cr_{1-\delta}Te$[2–8] are attractive owing to their rich magnetic properties, as well as inherent chemical and structural compatibility when forming heterostructures[9] with other topological systems, such as tetradymite-type topological insulators[10] or chalcogenide-based Dirac/Weyl semimetals[11]. Furthermore, the broken time-reversal symmetry

and spin-orbit coupling (SOC) offer unique opportunities for the interplay between spin configurations and reciprocal-space topology[12–14]. In this regard, ferromagnetic $Cr_2Te_3$ with strong perpendicular magnetic anisotropy (PMA) is an intriguing platform to host non-trivial topological physics, particularly for the high-quality thin films grown by molecular beam epitaxy (MBE)[15,16].

An important consequence of the band topology in $Cr_2Te_3$ is the Berry curvature[17,18] underlying the anomalous Hall effect (AHE)[19]. The

[1]Francis Bitter Magnet Laboratory, Plasma Science and Fusion Center, Massachusetts Institute of Technology, Cambridge, MA 02139, USA. [2]DEVCOM Army Research Laboratory, Adelphi, MD 20783, USA. [3]Department of Materials Science and Engineering, North Carolina State University, Raleigh, NC 27695, USA. [4]Department of Electrical and Computer Engineering, University of California, Riverside, CA 92521, USA. [5]Department of Physics, University of Maryland, College Park, MD 20742, USA. [6]Laboratory for Physical Sciences, College Park, MD 20740, USA. [7]Department of Materials Science and Engineering, Massachusetts Institute of Technology, Cambridge, MA 02139, USA. [8]Neutron Scattering Division, Neutron Sciences Directorate, Oak Ridge National Laboratory, Oak Ridge, TN 37831, USA. [9]Center for Nanophase Materials Sciences, Physical Science Directorate, Oak Ridge National Laboratory, Oak Ridge, TN 37831, USA. [10]Department of Physics, Georgetown University, Washington, DC 20057, USA. [11]Department of Physics, Northeastern University, Boston, MA 02115, USA. [12]Department of Physics, Massachusetts Institute of Technology, Cambridge, MA 02139, USA. ✉e-mail: chihang@mit.edu; ybou@mit.edu; moodera@mit.edu

intrinsic AHE is topological in nature and a hallmark of itinerant ferromagnets, which has also been observed in more exotic systems even without a net magnetization, such as spin liquids[20], antiferromagnets[21], and Weyl semimetals[22]. When SOC coexists with long-range magnetic order, the Berry curvature can be significantly influenced near avoided band crossings, rendering the system an incredibly rich playground combining topology and magnetism[23,24].

Here, we report the unique magnetotransport signatures of high-quality quasi-2D $Cr_2Te_3$ MBE-grown thin films governed by non-trivial band topologies. Via synergetic structural, magnetic, and transport measurements, together with first-principles simulations, we have uncovered novel Berry-curvature-induced magnetism featuring an extraordinary sign reversal of the AHE as we modulate the temperature and the strain for the thin films containing 3–24 unit cells (u.c.) on $Al_2O_3(0001)$ or $SrTiO_3(111)$ substrates. Moreover, a hump-shaped Hall feature emerges, most likely due to the presence of multiple magnetic layers/domains under different levels of interfacial strain. This work identifies pristine ferromagnetic $Cr_2Te_3$ thin films as a fascinating platform for further engineering topological effects, given their non-trivial Berry curvature physics.

## Results

### Atomic structure, interfaces, and strain

The crystalline structure of $Cr_2Te_3$ thin films is described first, followed by the development of strain at the substrate/film interface by the epitaxy. Bulk $Cr_2Te_3$ crystallizes in a three-dimensional (3D) lattice with

space group $P\bar{3}1c$ ($D_{3d}^2$, No.163), as shown in Fig. 1a–c, where each unit cell contains four vertically stacked hexagonal layers of Cr[25]. There are three symmetrically unique sites for Cr, labeled Cr1, Cr2, and Cr3, respectively: The Cr1 atoms are sparsely arranged in a weakly antiferromagnetic sublattice[26], while the Cr2/Cr3 atoms form ferromagnetic layers similar to those in $CrTe_2$[27]. Since the Cr1 sites are often only partially filled (Fig. 1i–m), $Cr_2Te_3$ behaves essentially as a quasi-2D magnet[28–30]. This quasi-2D nature of $Cr_2Te_3$ allows for high-quality, layer-by-layer epitaxial growth of $c$-oriented films on a variety of substrates. The hexagonal $c$ axis is the easy magnetic axis, leading to PMA for the films.

The sixfold in-plane (IP) symmetry is seen in the honeycombs visualized by atomic resolution scanning tunneling microscopy (STM, Fig. 1e) and scanning transmission electron microscopy (STEM, Fig. 1g) high-angle annular dark-field (HAADF) imaging, as well as in the reflection high-energy electron diffraction (RHEED, Supplementary Fig. 1) and X-ray diffraction (XRD, Supplementary Fig. 2) patterns. The sharp substrate/film interface is confirmed by the cross-sectional HAADF (Fig. 1i) and the corresponding integrated differential phase contrast (iDPC, Fig. 1j) images. The intrinsic random distribution of Cr atoms on the Cr1 sites is resolved in the enlarged view of the atoms in Fig. 1k–m, shown overlaid with red circles, while the overall chemical composition of the thin film is uniform within the resolution of energy dispersive X-ray spectroscopy (EDS, see Supplementary Fig. 3).

Figure 1f illustrates the basic sample architecture, where the strain in the $Cr_2Te_3$ thin films is governed by the interface with the substrate.

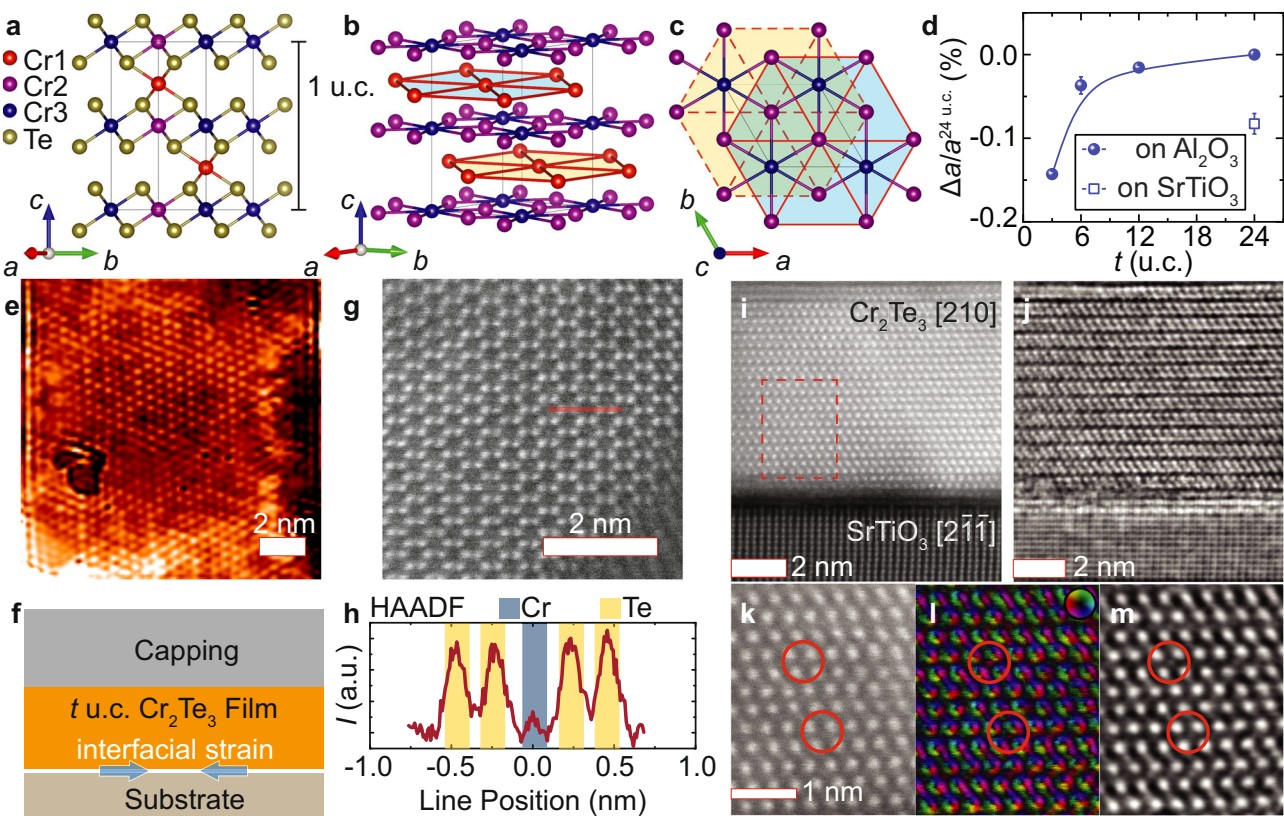

**Fig. 1 | Crystal structure of $Cr_2Te_3$ thin films. a** Atomistic structure of $Cr_2Te_3$ viewed along the crystallographic [210] direction. **b** Among the three Cr species, Cr1 (red) form sparse honeycombs that are stacked between those of Cr2/Cr3 (purple/blue) with sixfold in-plane symmetry (**c**). **d** Enhanced in-plane compressive strain at reduced thickness $t$, quantified by the relative change of the $a$ lattice parameter via XRD for $Cr_2Te_3$ grown on $Al_2O_3(0001)$ (solid) or $SrTiO_3(111)$ (open). **f** Schematic of the film stacks, where the interfacial strain plays a pivotal role in inducing extraordinary magnetic and transport phenomena. Atomically resolved STM morphology of a $13 \times 13$ nm² surface after removing Se capping (**e**) and planar

HAADF STEM image (**g**) of $Cr_2Te_3$ confirm the honeycomb-like Te lattice, where the HAADF intensity line scan reveals the Cr sites (**h**). **i–m** Cross-sectional images of $Cr_2Te_3$ films grown on $SrTiO_3(111)$. The HAADF (**i**) and iDPC (**j**) imaging along the [210] zone axis of $Cr_2Te_3$ illustrates the dominating Te–Cr2/Cr3–Te layers. The enlarged view (dashed box region in **i**) of HAADF (**k**), DPC (**l**), and iDPC (**m**) images identify the random distribution of the interlayer Cr1 (circles), which deviates from the ideal $Cr_2Te_3$ structure with full occupancy. The color wheel in the DPC image indicates the projected electric field direction.

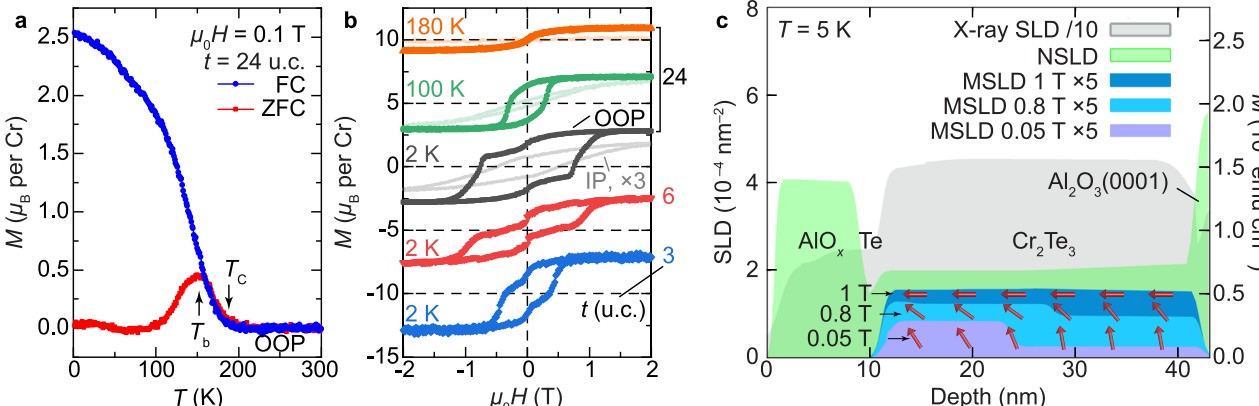

**Fig. 2 | Magnetic properties of Cr$_2$Te$_3$ thin films. a** Temperature dependence of the magnetization $M$ of a typical 24 u.c. Cr$_2$Te$_3$ film under the zero-field-cool (ZFC) and field-cool (FC) conditions with an out-of-plane (OOP) external magnetic field $\mu_0 H = 0.1$ T. The Curie ($T_C$) and blocking ($T_b$) temperatures are labeled by the arrows. **b** Field dependence of $M$ under OOP and in-plane (IP) configurations for $t = 24$ u.c. at selected temperatures (top three, black, green, and orange) and OOP $M(H)$ for $t = 6$ and 3 u.c. at 2 K (bottom two, red and blue). For clarity, the curves are vertically shifted, and the IP data are magnified by a factor of 3. **c** Depth profiles of PNR nuclear (NSLD), magnetic (MSLD, at IP fields of 1, 0.8, and 0.05 T, respectively) and X-ray scattering length densities (SLD) of 24 u.c. Cr$_2$Te$_3$ on Al$_2$O$_3$(0001) with Te/AlO$_x$ capping. The deduced spin configuration is schematically shown with red arrows overlaying with the MSLD profiles; the horizontal projection of the vectors corresponds to the IP $M$ determined by PNR.

Upon reducing the thickness $t$, films grown on Al$_2$O$_3$(0001) can develop an IP compressive strain up to −0.15%, as determined by XRD and summarized in Fig. 1d. A higher strain level can be sustained using SrTiO$_3$(111) substrates. Such control of strain is well suited for exploring interface-sensitive properties in Cr$_2$Te$_3$ thin films.

### Interface-driven magnetism revealed by PNR

The magnetic properties of Cr$_2$Te$_3$ thin films with selected thicknesses were assessed using vibrating sample magnetometry (VSM). Figure 2a shows the temperature dependence of the magnetization $M(T)$ for a $t = 24$ u.c. film on Al$_2$O$_3$(0001) substrate with an out-of-plane (OOP) applied magnetic field $\mu_0 H = 0.1$ T. Under the field-cool (FC) condition, $M(T)$ rises below the Curie temperature $T_C \sim 180$ K, reaching $M \sim 2.50 \mu_B$ (Bohr magneton) per Cr at 2 K in the 0.1 T field. The zero-FC (ZFC) scan, on the other hand, deviates from the FC curve below the blocking temperature $T_b$, signaling the freezing out of domains in a random direction in the absence of an aligning field.

As illustrated in Fig. 2b, Cr$_2$Te$_3$ favors PMA with coercive field $\mu_0 H_c = 0.76$ T and saturation magnetization $M_s \sim 2.83 \mu_B$ per Cr at 2 K for $t = 24$ u.c., whereas the IP measurements have weaker ferromagnetic hysteresis loops. The low-$T$ zero-field kink feature[31–33] in the OOP $M(H)$ becomes more prominent with reduced thickness (see Supplementary Fig. 4 for two-component analysis, as well as Supplementary Fig. 5 for additional data on $t = 6$ u.c.). The multistep hysteresis attests to the presence of varied layer-dependent magnetic anisotropies, despite the overall chemical and phase homogeneity of the films[34]. This is consistent with the interfacial strain-driven magnetic profiles revealed by the depth-sensitive polarized neutron reflectometry (PNR, Fig. 2c) as described below.

The PNR experiments, responsive to the IP magnetization, were carried out at chosen $T$ and $H$ on samples with $t = 24$ and 6 u.c. to uncover the impact of interfacial strain and the details of the stepwise hysteresis loops due to the interplay between anisotropy and the Zeeman energies in an applied external magnetic field[35,36]. The PNR spin asymmetry ratio $\mathrm{SA} = (R^+ - R^-)/(R^+ + R^-)$, measured as a function of the wave vector transfer $Q = 4\pi \sin(\theta)/\lambda$ with $R^+$ and $R^-$ being the reflectivity for the neutron spin parallel (+) or antiparallel (−) to the external field, evidently confirms the magnetization (Supplementary Fig. 6). By simultaneously refining PNR (measured at different $H$) and X-ray reflectivity (XRR, Supplementary Fig. 6) data, the depth profiles of nuclear (NSLD) and magnetic (MSLD) scattering length densities

(SLD) at $\mu_0 H = 1$ T, 0.8 T and 0.05 T for $t = 24$ u.c. were obtained and are shown in Fig. 2c. The uniform MSLD profile at the IP saturation field $\mu_0 H = 1$ T attests to the high quality of the magnetic Cr$_2$Te$_3$ film with well-defined interfaces of 0.5 nm roughness.

Remarkably, at reduced IP field $\mu_0 H = 0.8$ T and 0.05 T, $M$ develops a non-uniform depth-dependent profile with two distinct regions, possessing a lower (higher) IP magnetization value close to (away from) the substrate. Given that the NSLD depth profile of the Cr$_2$Te$_3$ layer is uniform and no changes are detected in the structure and chemical composition of the film, we attribute the reduced IP magnetization approaching the substrate to a canting of the magnetization vector towards the OOP direction (schematically drawn as red arrows in Fig. 2c). Since the OOP component of the magnetization vector is parallel to the momentum transfer $Q$, it is not responsive in PNR[37]. This is consistent with the observed PMA in the VSM measurements (Fig. 2b). These results collectively suggest that the more pronounced strain at the film/substrate interface leads to a higher OOP magnetic anisotropy and hence a lower measured IP MSLD.

The observed depth-dependent magnetization configuration is a result of the competition between the anisotropy energy and the Zeeman energy. Thus, under the IP configuration in the PNR experiments in Fig. 2c, with reduced IP external field, the Zeeman energy becomes insufficient to compete with the interfacial-strain-enhanced magnetic anisotropy term, giving rise to a restoration of a more OOP-oriented magnetization vector in the bottom layer. This magnetically soft layer is also responsible for the near-zero field kink in OOP $M(H)$ in Fig. 2b, where only a small OOP external field is needed for magnetic switching. To completely flip the magnetically harder top layer in the OOP configuration, though, a much higher coercive field is required (Supplementary Fig. 4). This is indeed consistent with the observation of a larger IP magnetization preserved in the top layer under reduced IP external field in Fig. 2c.

This scenario is further substantiated by the lower magnetization observed for $t = 6$ u.c. with stronger strain measured at 5 and 60 K under 1 T IP magnetic field (Supplementary Fig. 5d). The salient structural and magnetic features pave the way for an in-depth investigation of the magneto-transport responses in Cr$_2$Te$_3$ thin films.

### Strain-tunable AHE and sign reversal

The unusual Hall effects are the most outstanding properties of the Cr$_2$Te$_3$ thin films. The development of long-range magnetic ordering is

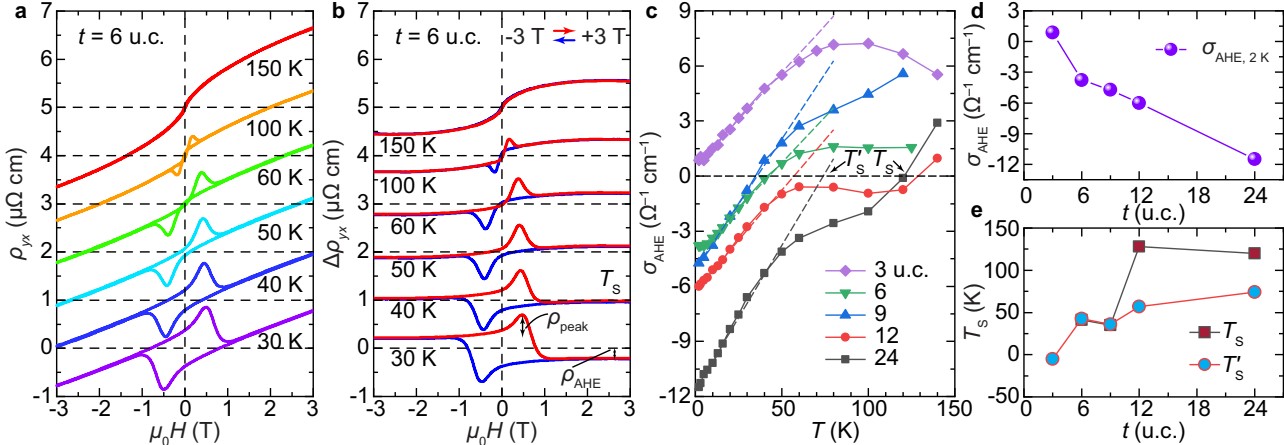

**Fig. 3 | The unconventional Hall effects in Cr$_2$Te$_3$ thin films. a** Magnetic field dependence of the Hall resistivity $\rho_{yx}(H)$ at selected $T$ for 6 u.c. Cr$_2$Te$_3$ on Al$_2$O$_3$(0001). **b** Hall traces $\Delta\rho_{yx}$ after removing the high-field ordinary Hall backgrounds. At $T_S \sim 40$ K, a sign change occurs in the anomalous Hall resistivity $\rho_{AHE}$, defined as the value of $\Delta\rho_{yx}$ when the system is fully magnetized under a positive $H$. Apart from the AHE hysteresis loop, additional hump-shaped features develop.

**c** Temperature dependence of the anomalous Hall conductivity $\sigma_{AHE}$ for $t = 3$–24 u.c. (symbols, where solid lines are guide for the eye and dashed lines are linear fit to low $T$ data). **d, e** Thickness dependence of $\sigma_{AHE}$ at 2 K (**d**), AHE sign reversal temperature $T_S$ (**e**), and the $T$-intercept of the linear AHE component at low $T$.

manifested in the AHE-induced hysteresis in the Hall resistivity

$$\rho_{yx}(H) = R_H H + R_S M, \tag{1}$$

in Fig. 3a (for more details on the transport parameters, see Supplementary Fig. 7). Here, $R_H$ characterizes the linear-in-$H$ ordinary Hall effect (OHE) that dominates at high $H$, and $R_S$ is the AHE coefficient denoting contribution from the underlying magnetic order.

By removing the linear OHE background in Fig. 3a, we now turn to the rich $T$ and $H$ dependences of the Hall traces $\Delta\rho_{yx}(H)$ and the unconventional AHE in the ferromagnetic regime in Fig. 3b. For $t = 6$ u.c., at $T \leq 30$ K, when fully magnetized under a positive $H$, the system produces a negative AHE signal $\rho_{AHE}$, i.e., $\Delta\rho_{yx}(H)$ loops around the origin in the opposite direction of that for the $M(H)$ hysteresis (Supplementary Fig. 5b). The $T$ dependence of the corresponding anomalous Hall conductivity $\sigma_{AHE} = \rho_{AHE}/(\rho_{AHE}^2 + \rho_{xx}^2)$, with $\rho_{xx}$ being the longitudinal electrical resistivity, is summarized in Fig. 3c. Upon rising $T$, $\rho_{AHE}$ changes sign at a transition temperature $T_S \sim 40$ K for $t = 6$ u.c. Note that the sign change signifies a compensation point at $T_S$ where $\rho_{AHE}$ or $\sigma_{AHE}$ traverses through zero while $M$ remains finite (see Supplementary Fig. 8b).

This is a highly intriguing transport behavior present in various members of the chromium telluride family of materials and related heterostructures[31,38–45], hitherto without consensus on a theoretical origin, yet strikingly similar to the anomaly in SrRuO$_3$ with a nontrivial band topology[23] and tunability of the Berry curvature via, e.g., epitaxial strain[46,47]. The sensitive interfacial strain dependence of the unique sign reversal behavior of the AHE in Cr$_2$Te$_3$ is illustrated in Fig. 3d. As evident in Fig. 3e, $T_S$ largely decreases upon increasing compressive strain at reduced $t$ (Fig. 1d). At $t = 3$ u.c., the strain is found to be sufficient to drive $\sigma_{AHE} > 0$ in the ground state, leading to the absence of a temperature-induced sign switching at finite $T$.

To elucidate the physical origin of the AHE sign reversal of Cr$_2$Te$_3$, we examined the Berry curvature $\Omega^z(\mathbf{k}) = \sum_n f_n \Omega_n^z(\mathbf{k})$ (Fig. 4a, summed over the occupied bands with $f_n$ the equilibrium Fermi-Dirac distribution function) based on the electronic band structure (Fig. 4b) obtained using density functional theory (DFT). As exemplified by the left inset of Fig. 4a, a significant spike feature develops in $\Omega^z(\mathbf{k})$, originating from the nearly degenerate SOC anti-crossing bands along the A-L $k$-path. The intrinsic AHE conductivity is evaluated by integrating

over the Brillouin zone (BZ)

$$\sigma_{AHE} = -\frac{e^2}{\hbar} \int_{BZ} \frac{d^3k}{(2\pi)^3} \Omega^z(\mathbf{k}), \tag{2}$$

where $e$ is the electron charge, and $\hbar$ is the reduced Planck's constant. The calculated $\sigma_{AHE} = -12.7\,\Omega^{-1}\,\text{cm}^{-1}$ at the Fermi level $\varepsilon_F$ for Cr$_2$Te$_3$ under equilibrium state (the black curve in Fig. 4c, see also Supplementary Fig. 9 for convergence test under different $k$-mesh), which is in excellent agreement with the experimental value of $-11.5\,\Omega^{-1}\,\text{cm}^{-1}$ for $t = 24$ u.c. It attests to the dominance of the intrinsic Berry phase mechanism, rather than the extrinsic side jump or skew scattering[19], as the primary origin of the observed AHE in Cr$_2$Te$_3$.

The calculation reveals a sensitive energy dependence of $\sigma_{AHE}$—not only the magnitude but also the sign change near $\varepsilon_F$. At finite $T$, due to the thermal broadening in $f_n$, the slight asymmetry of $\sigma_{AHE}$ above and below $\varepsilon_F$ may contribute to the observed AHE sign anomaly. Modeling of Berry curvature in strained cases in Fig. 4c further reveals that $\sigma_{AHE}$ at $\varepsilon_F$ changes sign under $-1\%$ compressive strain, substantiating that Berry physics underlies the observed strain-driven AHE sign reversal at base $T$ in Fig. 3d. The interface-induced two-component magnetic configuration in thicker films (as revealed by PNR in Fig. 2c), unambiguously traces the origin of the AHE results, that the transport in the more strongly strained bottom layer possesses the opposite sign from the rest of the layers, whose competition leads to an anomaly of the AHE sign at finite $T$. Thus collectively, these results demonstrate that epitaxial strain is the key reason for the sign change of AHE in Cr$_2$Te$_3$ films (Fig. 3d). The unique capability of achieving zero $\sigma_{AHE}$ or $\rho_{AHE}$ while maintaining nonzero $M$ in Cr$_2$Te$_3$ thin films, deviating from the classic Eq. (1), offers direct insight into the intrinsic AHE solely owing to the Berry curvature[45].

## Hump-shaped Hall peaks at the coercive field

Figure 3b also shows additional hump-shaped peaks on top of the otherwise square AHE hysteresis loop. The peaks are centered at the characteristic fields $H_{peak}$ that track well with the coercive fields $H_c$ determined from the magnetic measurements (Supplementary Fig. 8). These hump-shaped Hall peaks in our pristine Cr$_2$Te$_3$ are related to the presence of strain-modulated magnetic multilayer/domain structures with opposite signs of AHE (Fig. 5a), and not to the

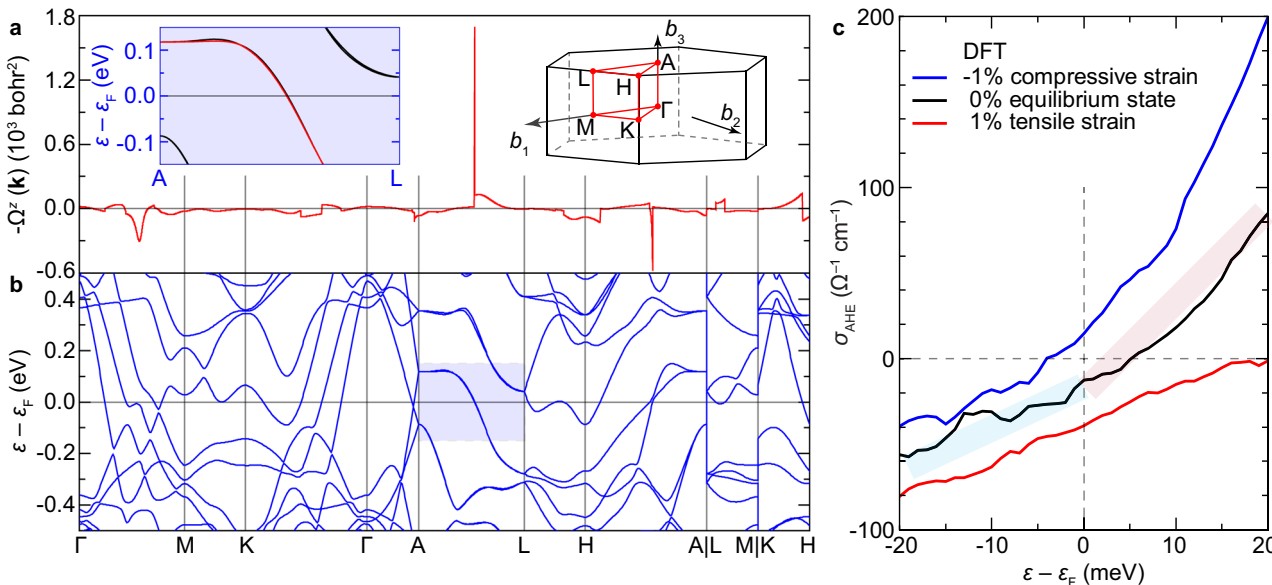

**Fig. 4 | Berry curvature and anomalous Hall conductivity in Cr₂Te₃.**
**a, b** Calculated Berry curvature $\Omega^z(\mathbf{k})$ (**a**) along the high symmetry $k$-paths in the Brillouin zone (right inset in **a**) and the corresponding electronic band structure (**b**). Left inset in **a**, nearly degenerate SOC anti-crossing bands contributing to the sharp

peak in $\Omega^z(\mathbf{k})$ along A–L. **c** Anomalous Hall conductivity $\sigma_{AHE}$ near the Fermi level $\varepsilon_F$, in equilibrium state (black), under compressive (blue) or tensile (red) strain conditions, respectively. The shades in **c** are guide for the eye showing the slight asymmetry of the energy dependence of $\sigma_{AHE}$ above and below $\varepsilon_F$.

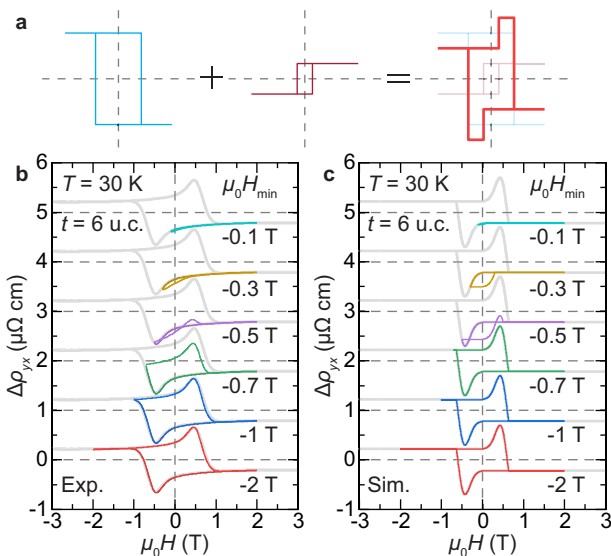

**Fig. 5 | Characteristics of hump-shaped Hall peaks in Cr₂Te₃. a** Simplified superposition of two AHE components with opposite sign and different coercive fields. **b**, **c** Minor loop scans of 6 u.c. Cr₂Te₃ film on Al₂O₃(0001) at $T = 30$ K, first fully magnetized at $\mu_0 H = +3$ T (complete loop shown in gray as guide for the eye) and then swept back and forth between +2 T and selected $\mu_0 H_{min}$. The experimental minor loops in **b** are qualitatively reproduced in **c** using simulations that underscore the significance of strain-driven multilayer/domain features and the sign reversal in $\rho_{AHE}$.

skyrmion-induced topological Hall effect as postulated in various heterostructures[38–40,42–44,48].

To better understand the mechanism(s) underlying the Hall peaks observed in $\Delta\rho_{yx}(H)$, minor loop experiments were carried out at $T = 30$ K and are shown in Fig. 5b. For each scan, the loop starts from a well-defined initial state that is fully magnetized under a positive $H$, which is then swept towards a negative $H_{min}$ around $-H_{peak}$ and scanned back to the initial positive $H$. The minor loops are hysteretic, where

the emergence of the Hall peak with positive $H$ depends on whether $H_{min}$ surpasses $-H_{peak}$.

The two-component origin of the Hall anomaly peaks in transport corroborates with the magnetic structure revealed in Fig. 2c and Supplementary Fig. 4, which is well explained by the distribution of magnetic multilayers/domains, modulated via interfacial strain with $T$-dependent $H_c$, using[49]

$$\Delta\rho_{yx}(H) = \int_0^\infty \widetilde{\rho}_{AHE}(T')\left\{2H_{Heav}\left[H - \widetilde{H}_c(T')\right] - 1\right\}G(T')dT'. \quad (3)$$

Here $\widetilde{\rho}_{AHE}(T')$ and $\widetilde{H}_c(T')$ are functionals based on experimental $\rho_{AHE}$ and $H_c$ (Supplementary Fig. 8), $H_{Heav}(x)$ is the Heaviside function approximating the switching of $M$, and the Gaussian distribution

$$G(T') = \frac{1}{\sqrt{2\pi T_\sigma^2}}\exp\left[-\frac{(T'-T)^2}{2T_\sigma^2}\right], \quad (4)$$

characterizes the strain-driven distribution of magnetic layers/domains with varying $T_S$ by assuming an effective temperature spreading factor $T_\sigma$. As compared in Fig. 5c, the numerical simulation indeed qualitatively well reproduces the behavior of minor loops. The observed AHE sign change and the emergence of hump-shaped Hall features are also present in films grown on SrTiO₃(111) (Supplementary Fig. 10). The quality of the substrate/film interface plays a pivotal role in materializing this exquisite tunability of the Berry curvature in Cr₂Te₃ films.

In summary, we have discovered unusual strain-modulated Berry curvature-driven effects in the anomalous Hall transport of Cr₂Te₃ thin films. We report on the growth, detailed magnetic, and transport properties of pristine Cr₂Te₃ MBE thin films deposited on Al₂O₃(0001) and SrTiO₃(111) substrates. A striking sign reversal in the anomalous Hall resistivity, accompanied by a finite magnetization, has been observed and theoretically modeled, revealing the relevance of the nontrivial Berry curvature physics. This unique sign reversal, coupled with the intrinsic strain-induced magnetic multilayer/domain structure in the material, underpins a hump-shaped Hall feature in Cr₂Te₃ thin

films. The Berry curvature effect is observed in this case due to the high quality of the substrate/film interface, which is further tunable via different levels of strain from varying film thickness and/or choice of substrates. Our comprehensive experimental and theoretical investigations have established the strain-sensitive $Cr_2Te_3$ and physics-rich broader $Cr_{1-\delta}Te$ family of materials to host tunable topological effects related to the intrinsic Berry curvature, thereby providing new perspectives in the field of topological electronics.

## Methods

### Sample growth

The growth of $Cr_2Te_3$ thin films, with nominal $t$ ranging from 3 to 24 u.c., was carried out in an MBE system under an ultrahigh-vacuum (UHV) environment of $10^{-10} - 10^{-9}$ Torr. Insulating $Al_2O_3(0001)$ was primarily used as a substrate, whose surface quality was insured by ex situ chemical and thermal cleaning and in situ outgassing at 800 °C for 30 min. When using $SrTiO_3(111)$, the insulating substrates were first annealed at 930 °C for 3 h in a tube furnace under a flowing oxygen environment to achieve a passivated surface with atomic flatness and then in situ outgassed at 580 °C for 30 min. After surface preparation, the substrate temperature was lowered to 230 °C for film growth, allowing enough surface mobility for the epitaxial crystallization of the desired phase of $Cr_2Te_3$. High-purity (5 N) Cr was evaporated from an e-beam source, while Te was thermally co-evaporated from a Knudsen effusion cell adjusted to maintain a typical Cr:Te flux ratio of 1:10 and a growth rate of approximately 0.005 nm s$^{-1}$. The epitaxial growth process was monitored by in situ RHEED (see Supplementary Fig. 1) operated at 15 kV. The as-grown films were in situ annealed at the growth temperature for 30 min and naturally cooled to room temperature. For ex situ characterizations, films were protected by in situ capping with Te (2 nm) and $AlO_x$ (10 nm) or Se (20 nm) for later removal for STM measurements. The schematic of the film stack is illustrated in Fig. 1f.

### Structural characterizations

The XRD patterns were obtained using a parallel beam of Cu $K_{\alpha 1}$ radiation with wavelength $\lambda = 0.15406$ nm in a Rigaku SmartLab system. The $2\theta$ (for OOP measurement) and/or $2\theta_\chi$ (for IP configuration) scan angles were between 10° and 120° with a typical step size of 0.05°. XRR measurements were performed at the Center for Nanophase Materials Sciences (CNMS), Oak Ridge National Laboratory, on a PANalytical X'Pert Pro MRD equipped with a hybrid monochromator and Xe proportional counter. For the XRR measurements, the X-ray beam was generated at 45 kV/40 mA, and the X-ray beam wavelength after the hybrid mirror was $\lambda = 0.15406$ nm (Cu $K_{\alpha 1}$ radiation). To facilitate electron microscopy, plan view samples were deposited on $Si_3N_4$ TEM grids with thin $Sb_2Te_3$ buffer while cross-sectional samples were prepared using the focused ion beam (FIB) lift-out method on a Thermo Scientific FEI Quanta 3D dual beam system. STEM imaging was carried out on a Thermo Scientific FEI Titan aberration-corrected system operated at 200 kV. A semi-convergence angle of 17.9 mrad was used. DPC and iDPC images were recorded using a segmented detector. For the 3 u.c. sample, STEM images were acquired with a Themis Z G3 instrument provided by Thermo Fischer Scientific at 200 kV with a beam current of 40 pA and a convergence semi-angle of 20 mrad.

### Scanning tunneling microscopy

STM experiments were performed at the Laboratory for Physical Sciences using a home-built low-temperature scanning tunneling microscope[50] controlled by a Topometrix digital feedback electronic control unit. Samples were loaded into a UHV environment with a base pressure of $5 \times 10^{-10}$ Torr and heated in front of a residual gas analyzer to verify the removal of the Se capping layer before being transferred to the microscope at 77 K. Scans were performed with an electro-chemically etched tungsten tip and differential spectroscopy data

were extracted via a Stanford Research Systems SR830 lock-in amplifier.

### Polarized neutron reflectometry

PNR is a highly penetrating depth-sensitive technique to probe the chemical and magnetic depth profiles with a resolution of 0.5 nm. The depth profiles of the NSLD and MSLD correspond to the depth profile of the chemical and IP magnetization vector distributions on the atomic scale, respectively[51–53]. Based on these neutron scattering merits, PNR serves as a powerful technique to simultaneously and nondestructively characterize the chemical and magnetic nature of buried interfaces[54]. PNR experiments were performed on the Magnetism Reflectometer at the Spallation Neutron Source at Oak Ridge National Laboratory[55–57], using neutrons with wavelengths $\lambda$ in a band of 0.2–0.8 nm and a high polarization of 98.5–99%. Measurements were conducted in a closed-cycle refrigerator (Advanced Research System) equipped with a 1.15 T Bruker electromagnet. Using the time-of-flight method, a collimated polychromatic beam of polarized neutrons with the wavelength band $\Delta\lambda$ impinges on the film at a grazing angle $\theta$, interacting with atomic nuclei and the spins of unpaired electrons. The reflected intensity $R^+$ and $R^-$ are measured as a function of the wave vector transfer, $Q = 4\pi \sin(\theta)/\lambda$, with the neutron spin parallel (+) or antiparallel (−), respectively, to the applied field. To separate the nuclear from the magnetic scattering, the spin asymmetry ratio $SA = (R^+ − R^-)/(R^+ + R^-)$ is calculated, for which $SA = 0$ designating no magnetic moment in the system. Being electrically neutral, spin-polarized neutrons penetrate the entire multilayer structures and probe the magnetic and structural composition of the film and buried interfaces down to the substrate.

### Transport and magnetic measurements

Electrical transport measurements as a function of temperature and field were performed in the temperature range of 2–300 K in a Quantum Design Physical Property Measurement System (PPMS) equipped with a 9 T superconducting magnet. A typical ac current ($I_x$) of 5 μA was injected into the Hall bar (-0.3 × 1.0 mm² for hand-scratched or 10 × 30 μm² for e-beam patterned) residing in the crystallographic $a$−$b$ plane, while longitudinal ($V_x$) and transverse ($V_y$) voltages were simultaneously monitored using a lock-in technique. VSM was used to characterize the magnetization, where linear diamagnetic backgrounds from sample holders/substrates were subtracted to obtain $M(H)$ and $M(T)$.

### Theoretical calculations

First-principles calculations were performed using the Quantum Espresso packages[58]. The generalized gradient approximation with the Perdew−Burke−Ernzerhof parameterization (GGA-PBE) was used as the exchange-correlation functional[59]. An energy cutoff of 40 Ry and a $6 \times 6 \times 4$ Γ-centered $k$-mesh were applied for the relaxation calculation. The crystal structure of $Cr_2Te_3$ was fully optimized until the force on each atom was smaller than 0.05 eV nm$^{-1}$. The optimized lattice constants of bulk $Cr_2Te_3$ are $a = b = 0.6799$ nm and $c = 1.2022$ nm. For the self-consistent field calculation, SOC was included, and a higher $12 \times 12 \times 8$ $k$-mesh was used. The magnetization was set along the $z$-axis. The resulting absolute magnetic moments of the Cr atoms are 3.08, 2.99, and 3.06 $\mu_B$ for Cr1, Cr2, and Cr3, respectively. For the Berry curvature and anomalous Hall conductivity calculations, Wannier90 packages were used[60]. Maximally localized Wannier functions, including both Cr $d$-orbitals and Te $p$-orbitals were employed to reproduce the DFT-calculated band structure with SOC.

## Data availability

The data that support the findings of this study are available from the corresponding authors upon reasonable request.

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

## Acknowledgements

The work at MIT was supported by the Army Research Office (W911NF-20-2-0061 and DURIP W911NF-20-1-0074), National Science Foundation (NSF-DMR 1700137 and 2218550), and Office of Naval Research (N00014-20-1-2306). H.C. was sponsored by the Army Research Laboratory (ARL) under Cooperative Agreement Number W911NF-19-2-0015. Y.O. and J.S.M. thank the Center for Integrated Quantum Materials (NSF-DMR 1231319) for financial support. D.H. thanks support from NSF grant DMR-1905662 and the Air Force Office of Scientific Research award FA9550-20-1-0247. T.B.E. is partially supported by NSF under Grant No. DGE 1633587. A.C.F. is supported by the MIT-IBM Watson AI Lab. The electron microscopy was performed at the Analytical Instrumentation Facility (AIF) at North Carolina State University, which is supported by the State of North Carolina and NSF (Award No. ECCS-2025064). The AIF is a member of the North Carolina Research Triangle Nanotechnology Network (RTNN), a site in the National Nanotechnology Coordinated Infrastructure (NNCI). This work was carried out with the use of facilities and instrumentation supported by NSF through the Massachusetts Institute of Technology Materials Research Science and Engineering Center DMR—1419807. This work was carried out in part through the use of MIT.nano's facilities. This research used resources at the Spallation Neutron Source, a Department of Energy Office of Science User Facility operated by the Oak Ridge National Laboratory. XRR measurements were conducted at the Center for Nanophase Materials Sciences (CNMS), which is a DOE Office of Science User Facility. STM measurements utilized the facilities and resources of the Laboratory for Physical Sciences. The DFT work was supported in part by the ARL Research Associateship Program (RAP) Cooperative Agreement Number W911NF-16-2-0008 and used STAMPEDE2 at TACC through allocation DMR130081 from the Advanced Cyberinfrastructure Coordination Ecosystem: Services & Support (ACCESS) program, which is supported by NSF grants #2138259, #2138286, #2138307, #2137603, and #2138296.

## Author contributions

H.C., Y.O., and J.S.M. conceived and designed the research. H.C. grew the films with assistance from Y.O., J.S.M., C.R., and P.J.T., carried out magnetization measurements with D.H., collected and analyzed transport data on macroscopic initial Hall bars as well as microdevices fabricated by O.A.V. T.B.E., W.G., A.C.F., and F.M.R. examined the microstructure using STEM. J.M., M.D., and R.E.B. characterized the surface using STM. J.K. performed XRR measurements, V.L. and H.A. conducted PNR experiments, and V.L. analyzed XRR and PNR data. S.K., Y.L., M.R.N., G.Y., and R.K.L. conducted first-principles calculations while A.T.G., G.J.d.C., and P.A.F. simulated micromagnetic responses. H.C., D.H., and J.S.M. wrote the paper, with contributions from all authors.

## Competing interests

The authors declare no competing interests.
