## [Peer Review File · Nature Communications]

Reviewers' Comments:

Reviewer #1:

Remarks to the Author:

This manuscript reports a sign reversal anomalous Hall resistivity of strained Cr₂Te₃ thin films, including the theoretically modeled Berry curvature simulations. Quasi-two-dimensional structure leads extraordinary topological effect on Cr₂Te₃. The study shows a good impression and interesting results to understand a unique sign reversal anomalous Hall resistivity. However, the manuscript lacks several essential data to corroborate roles of the strain-related issues. If this manuscript could be further considered to publish, following questions must be addressed.

1) In the Figure 1d, the error bar of a compressive strain for 6 u.c. on Al₂O₃ is too higher than other samples, enough to cover whole strain ranges, and authors focused a thickness of 6 u.c. to analyze magneto-electrical transports. Is there any reason? Based on the figure 1d, even though a line is a guide for eyes, 8-9 u.c. of Cr₂Te₃ on Al₂O₃ should be included to compare 24 u.c. Cr₂Te₃ on SrTiO₃ (Figure S8). As a perspective for the strain, both samples make good agreements each other in the series of analysis, if the role of strain is critical to govern the reversal anomalous Hall effect.

2) In the Figure 2b, H_c of 3 u.c. dropped to half from 1 T, while M of 3 and 24 u.c. is almost same. M of 6 u.c. still contains diagonal background, it is actually ~80 % of other samples. Interfacial strain could affect those changes, but there is no explain relevant to the strain displayed in Figure 1d.

3) In the Figure 2c, NSLD and MSLD are clear and impressive. But X-ray SLD shows a small bump from a top of Cr₂Te₃. EDAX line profile would be helpful to exclude a possibility of an artifact for stoichiometry/phase different Cr_xTe_y compounds.

4) T_s is mentioned in the manuscript only. Please add T_s to Figure 3c for the readers if possible.

5) In p. 2, line 44, the authors described a compressive strain increasing up to -0.15 %. It is a same value for 3 and 6 u.c. samples. However, not only strain but also thicknesses affect anomalous resistivity, as shown in Figure 3c. Authors claimed the AHE is sensitively governed by the interfacial strain (p. 4, line 3) and strain at 3 u.c. is to be sufficient in driving r_{AHE}>0 (p. 4, line 6). It is one of key points for a thickness having r_{AHE}>0. Why 6 u.c. is not sufficient in driving r_{AHE}>0 even same strain applied situation?

6) Roughly, strain of 6, 12 and 24 are -0.15 %, -0.05 %, and almost 0 %, respectively. 9 u.c. sample could be -0.1 %. Those are good numbers to show a trend of T_s. The authors presented an existence of 12 u.c. sample in the Figure 1d. Does T_s as a function of thickness agree with Figure 1d except 3 u.c.?

7) A lattice mismatch between substrate and growing film leads the strain at the interface. And strain relaxation occurs in the over-layers gradually while growth. It means that the presence of a strain at interface is always existed with similar amount when films grown on specific substrate. In this regards, different strain for 24 u.c. on Al₂O₃ and SrTiO₃ is reasonable (in the Figure 1d). However, it is not mean that an obtained compressive strain is equivalent to the interface strain. As shown in the Figure 1j, mono/bi-layer of Te formed on STO and defective 1 u.c. Cr₂Te₃ grew. After that, above films were grown well. Authors should present a study of a strained interface and a structure of top-most area via STEM and FFT analysis.

8) The hump-shape Hall peaks in AHE can be understood by a superposition of two AHE hysteresis loops from strained interface and relaxed overlayers as my aforementioned concern. 2 AHE fitting for different thickness at 30 K should be presented. It is possible that one AHE is proportional to the thickness while opposite sign of AHE remains regardless of thickness.

Reviewer #2:

Remarks to the Author:

The author performed systematic study on cr₂te₃ thin films on two different substrates, revealing an interesting strain-tunable sign reversal effect of AHE, together with topological Hall effect. Overall, the epitaxy and the characterization are of high quality, especially for the ultra-thin sample(6 unit cells) of strain, a high contrast RHEED pattern was observed clearly. However, I didn't observe the RHEED coming from 3 unit cell sample, as well as the TEM characterization for the sample of 3 unit cell. Since the disappearance of sign change are robustly observed in these sample, I suggest the authors provide the structure analysis on these 3 unit cell sample, before

the acceptance of this article's major claim.

Reviewer #3:

Remarks to the Author:

Report of the manuscript entitled as "Strain-tunable Berry curvature in quasi-two-dimensional Chromium Telluride" by Hang Chi et al.

Transport behaviors in 2 dimensional materials are attracting considerable attention. Among many different material systems, binary compound of composition $\text{Cr}_{(1-\delta)}\text{Te}$ is particularly interesting for bridging the band topology and magnetism. On this material system, many intriguing behaviors have been observed, including the giant anomalous Hall effect and temperature-dependent sign change of anomalous Hall effect. On this very topic, through synthesizing MBE grown Cr_2Te_3 thin films, authors studied the nontrivial topology and their contribution to the anomalous Hall effect. In particular, authors identified a hump-like Hall feature, together with a sign-reversal of AHE, which were attributed to the occurrence of spin textures with non-trivial topology.

While the data presented in the manuscript is high-quality, the conclusion is, however, not beyond the those published in the past. In fact, the main results in the present manuscript can be found from literatures. Considering the current development and the novelty of their results, I see no reasons that this manuscript could be published in journals such as Nature Communications. Below I outline my main concerns.

1: High-quality $\text{Cr}_{(1-\delta)}\text{Te}$ films have been synthesized by MBE. From this aspect, the material system is lacking of sufficient novelty.

2: Bump and dip features associated with the intriguing anomalous Hall effect in CrTe films have also been frequently reported. Their possible connection with both the real-space and reciprocal space topology has been suggested and discussed. Thus, this manuscript did not advance the current understanding on the intriguing anomalous Hall effect in CrTe films.

3: The sign change of anomalous Hall effect in CrTe films has also been previous reported.

While the experimental part of the current manuscript is probably high-quality and systematic, the interpretation is lack of sufficient experimental evidence, especially the identification of nontrivial real space topological spin textures and band topology. In this sense, I don't believe it meets the publishing criteria of Nature Comm.

A: This manuscript needs a careful proof reading.

B: Authors mentioned that "sizable spin-orbit interaction" and "Dzyaloshinskii-Moriya interaction" can be found in the current material system. However, no estimation of both parameters is attempted.

C: While I appreciate the detailed structural characterization, I am not quite sure if they serve for the purpose of "strain-tunable". In fact, similar behaviors can be observed on both substrates, and authors failed to tune their observation by any other means.

D: Depending on the stoichiometry, $\text{Cr}_{(1-\delta)}\text{Te}$ compounds contain rich phases. While authors claimed the composition of their thin films is Cr_2Te_3 , the two-step magnetometry data suggest their thin films is not single phase. In fact, shapes of hysteresis loops are lacking of any meaning discussions. Competing contributions from different structural phases could substantially complicate the data analyses. The discussion of blocking temperature needs to be re-examed, which may well suggest the structural inhomogeneity in the system. Thus, from the experimental data, I am not convinced their bump and dip features contain a topological origin.

E: Neither an experimental reveal of nontrivial band topology by using ARPES, nor an experimental identification of real-space spin topology are being made. From an experimental perspective, very limited new messages are delivered from this manuscript.

F: How to estimate the spin polarization of CrTe film? Did they attempt to experimentally measure the possible existence of topological spin textures, since the size of which could be affected by temperature.

In summary, on a rather well-studied material system, authors revisited the intriguing anomalous Hall effect in CrTe films, and attributed their observations to the combined contribution from band structure topology and real-space spin topology. Their conclusions are however, lacking of sufficient experimental evidences. I am against its publication.

REVIEWER COMMENTS

Reviewer #1 (Remarks to the Author):

This manuscript reports a sign reversal anomalous Hall resistivity of strained Cr₂Te₃ thin films, including the theoretically modeled Berry curvature simulations. Quasi-two-dimensional structure leads extraordinary topological effect on Cr₂Te₃. The study shows a good impression and interesting results to understand a unique sign reversal anomalous Hall resistivity. However, the manuscript lacks several essential data to corroborate roles of the strain-related issues. If this manuscript could be further considered to publish, following questions must be addressed.

Response: We thank the referee for the positive evaluation of our work. We have revised the manuscript extensively, following the referee's constructive suggestions. Notably, the new DFT result of strain-induced sign reversal in the anomalous Hall conductivity (new **Fig. 4c**), first of its kind in the literature to the best of our knowledge, corroborates well our experimental observation that strain plays an essential role in the anomalous Hall phenomenology in Cr₂Te₃. We performed additional PNR experiments at several values of magnetic field, showing the evolution of the magnetization depth profile. We also added additional PNR data for 6 u.c..

1) *In the Figure 1d, the error bar of a compressive strain for 6 u.c. on Al₂O₃ is too higher than other samples, enough to cover whole strain ranges, and authors focused a thickness of 6 u.c. to analyze magneto-electrical transports. Is there any reason? Based on the figure 1d, even though a line is a guide for eyes, 8-9 u.c. of Cr₂Te₃ on Al₂O₃ should be included to compare 24 u.c. Cr₂Te₃ on SrTiO₃ (Figure S8). As a perspective for the strain, both samples make good agreements each other in the series of analysis, if the role of strain is critical to govern the reversal anomalous Hall effect.*

Response: We thank the referee for the instructive comment and agree that the original error bar for 6 u.c. was overestimated as it included data from regions in (006,12), where the peaks are *not* well defined for 6 and 3 u.c. due to low counts. We have now improved the error bar in the new **Fig. 1d** using a Gaussian fit to the XRD (004,8) peaks. We note that the thickness dependent systematic evolution of the lattice parameter/strain manifests in the shift of the (008) peaks in the original Fig. S2b. Transport data for additional films are now also included in the new **Fig. 3c-e** to complement the analysis, substantiating the important role of strain.

2) *In the Figure 2b, H_c of 3 u.c. dropped to half from 1 T, while M of 3 and 24 u.c. is almost same. M of 6 u.c. still contains diagonal background, it is actually ~80 % of other samples. Interfacial strain could affect those changes, but there is no explain relevant to the strain displayed in Figure 1d.*

Response: We thank the referee for the careful examination of our data. The diamagnetic background was removed by linear approximation of the high-field $M(H)$ data. Such a standard procedure indeed correctly restores the saturated magnetization $M_s = 2.57 \mu_B$ per Cr for 6 u.c., which is ~ 90% of that for 24 u.c.. An additional discussion is now included pertaining to the thickness dependent evolution of the magnetic properties, which suggests two component contribution – one from the layer farthest from the substrate (lowest strain) exhibiting a higher H_c .

value, the other one from the film sublayer close to the substrate (higher strain), leading to a lower H_c .

In order to unravel the thickness dependent evolution of the magnetic properties we performed additional PNR measurements at different magnetic fields included now in revised **Fig. 2c**. The data show that with reducing magnetic field the magnetization M is not uniform and shows a lower M close to the substrate where the strain is higher and a higher M further from the substrate. Given that PNR is sensitive to the in-plane magnetization, these results directly confirm the scenario of two component distribution. In this case the region with a higher OOP anisotropy will show a lower MSLD measured with in-plane magnetic field.

In addition, we included in **Fig. S4d** PNR data for 6 u.c. sample, measured with 1 T at 5 K and 60 K, which shows a lower in-plane M compared to 24 u.c. sample due to a stronger OOP anisotropy. Note that PNR measures the absolute value of magnetization vector, where the in-plane M measured by PNR is shown in the 2nd axis, using conversion $1 \text{ emu/cc} = 2.9 \times 10^{-9} \text{ \AA}^{-2}$.

The relevant revised text on Pages 4-5 reads:

“Depth-sensitive polarized neutron reflectometry (PNR) measurements, responsive to the IP magnetization, were carried out **at chosen T and H on samples with $t = 24$ and 6 u.c., in order to uncover the impact of interfacial strain**. The PNR spin asymmetry ratio $SA = (R^+ - R^-)/(R^+ + R^-)$, measured as a function of **wave vector transfer $Q = 4\pi\sin(\theta)/\lambda$ with R^+ and R^- being the reflectivity for the neutron spin parallel (+) or antiparallel (-) to the external field, evidently confirms the magnetization (Supplementary Fig. 5)**. By simultaneously refining PNR and X-ray reflectivity (XRR, Supplementary Fig. 5) data, the depth profiles of nuclear (NSLD) and magnetic (MSLD) scattering length densities **(SLD) at $\mu_0 H_c = 1 \text{ T}$, 0.8 T and 0.05 T for $t = 24 \text{ u.c.}$** were obtained and shown in **Fig. 2c**. The **uniform MSLD profile at the IP saturation field $\mu_0 H = 1 \text{ T}$ attests to the high quality of the magnetic Cr_2Te_3 layer with well-defined interfaces of 0.5 nm. Remarkably, at a reduced IP field $\mu_0 H = 0.8 \text{ T}$ and 0.05 T, M develops a non-uniform depth-dependent magnetization profile with two components, revealing a lower (higher) value close to (away from) the substrate. Since PNR is sensitive to IP magnetization, these results collectively suggest that more pronounced strain at the interface leads to a higher OOP magnetic anisotropy and hence a lower measured IP MSLD. This scenario is further substantiated by the lower M observed for $t = 6 \text{ u.c.}$ with stronger strain measured at 5 K and 60 K with 1 T IP magnetic field (Supplementary Fig. 4d)**. The salient structural and magnetic features pave the way for **an** in-depth investigation of the magneto-transport responses in Cr_2Te_3 thin films.”

3) *In the Figure 2c, NSLD and MSLD are clear and impressive. But X-ray SLD shows a small bump from a top of Cr_2Te_3 . EDAX line profile would be helpful to exclude a possibility of an artifact for stoichiometry/phase different Cr_xTey compounds.*

Response: We thank the referee's suggestion. We cross-checked the analysis of the XRR data, re-fitted the data which showed a negligible sensitivity to this small bump. The new fit is now presented in revised **Fig. 2c**. STEM EDS distribution is also provided in new **Fig. S3** with a uniform elemental distribution, suggesting that an artifact for stoichiometry/phase variation at the top of the film stack may be ruled out.

The relevant revised text on Page 3 reads

“The intrinsic random distribution of Cr atoms on the Cr1 sites is resolved in the enlarged view of the atoms in **Figs. 1k-m**, shown overlaid with red circles, while the overall chemical composition of the thin film is uniform within the resolution of energy dispersive X-ray spectroscopy (EDS, see Supplementary Fig. 3).”

4) T_s is mentioned in the manuscript only. Please add T_s to Figure 3c for the readers if possible.

Response: We thank the referee’s instruction and have added T_s to new **Fig. 3**.

5) In p. 2, line 44, the authors described a compressive strain increasing up to -0.15 %. It is a same value for 3 and 6 u.c. samples. However, not only strain but also thicknesses affect anomalous resistivity, as shown in Figure 3c. Authors claimed the AHE is sensitively governed by the interfacial strain (p. 4, line 3) and strain at 3 u.c. is to be sufficient in driving $r_{AHE} > 0$ (p. 4, line 6). It is one of key points for a thickness having $r_{AHE} > 0$. Why 6 u.c. is not sufficient in driving $r_{AHE} > 0$ even same strain applied situation?

Response: We thank the referee’s observation. As revealed in the new **Fig. 1d**, 3 u.c. actually possesses a larger strain than 6 u.c., confirming its chief role in governing the anomalous Hall behavior. This effect is now also established theoretically by DFT calculation presented in new **Fig. 4c**, where a sign reversal under compressive strain is discovered.

6) Roughly, strain of 6, 12 and 24 are -0.15 %, -0.05 %, and almost 0 %, respectively. 9 u.c. sample could be -0.1 %. Those are good numbers to show a trend of T_s . The authors presented an existence of 12 u.c. sample in the Figure 1d. Does T_s as a function of thickness agree with Figure 1d except 3 u.c.?

Response: We thank the referee’s advice and have now measured 12 and 9 u.c. samples and compiled data in new **Fig. 3c-e**. T_s displays a largely increasing trend upon increasing t , in reasonable agreement with the overall decreasing level strain in **Fig. 1d**.

7) A lattice mismatch between substrate and growing film leads the strain at the interface. And strain relaxation occurs in the over-layers gradationally while growth. It means that the presence of a strain at interface is always existed with similar amount when films grown on specific substrate. In this regards, different strain for 24 u.c. on Al₂O₃ and SrTiO₃ is reasonable (in the Figure 1d). However, it is not mean that an obtained compressive strain is equivalent to the interface strain. As shown in the Figure 1j, mono/bi-layer of Te formed on STO and defective 1 u.c. Cr₂Te₃ grew. After that, above films were grown well. Authors should present a study of a strained interface and a structure of top-most area via STEM and FFT analysis.

Response: We thank the referee’s insight. The overall level of compressive strain of the entire film stack (derived from XRD in **Fig. 1d**) was utilized to conceptualize the primary origin underlying the AHE sign reversal, while the authors fully agree with the referee’s recognition of the ubiquitous defects and structural relaxation from the film/substrate interface.

The FFT analysis is shown below, for new STEM image (**Fig. S3b**) covering both strained interface (Bottom) and top-most area (Top). The FFT reveals contrast between Bottom and Top to a limited extent, largely due to the small film thickness involved.

To better capture the effects pertaining to the strained interface, the new PNR results are better suited to offer a more wholistic assessment (see **Fig. 2c** and response to Q2 above).

8) *The hump-shape Hall peaks in AHE can be understood by a superposition of two AHE hysteresis loops from strained interface and relaxed overlayers as my aforementioned concern. 2 AHE fitting for different thickness at 30 K should be presented. It is possible that one AHE is proportional to the thickness while opposite sign of AHE remains regardless of thickness.*

Response: We thank the referee for the advice – and note that our simulated result is exactly the multiple AHE channel fitting. It followed the approach of Kan *et al.* [*Phys. Rev. B* **98**, 180408 (2018)], where the simplified 2 AHE fitting is improved by a more realistic Gaussian distribution of multiple channels. The **Fig. 5** has now been revised to highlight the 2 AHE fitting.

The relevant revised text on Page 8 reads:

“**Figure 3b** also shows additional hump-shaped peaks on top of the otherwise square AHE hysteresis loop. The peaks are centered at the characteristic fields H_{peak} that track well with the coercive fields H_c determined from the magnetic measurements (Supplementary Fig. 7). These hump-shaped Hall peaks in our pristine Cr_2Te_3 are related to the presence of strain-modulated magnetic multidomain structures with opposite signs of AHE (**Fig. 5a**), rather than the skyrmion-induced topological Hall effect as postulated in various heterostructures³⁵⁻⁴¹. To better understand the mechanism(s) underlying the Hall peaks observed in $\Delta\rho_{yx}(H)$, minor loop experiments were carried out at $T = 30$ K and are shown in **Fig. 5b**. For each scan, the loop starts from a well-defined initial state that is fully magnetized under a positive H , which is then swept towards a negative H_{min} around $-H_{\text{peak}}$ and scanned back to the initial positive H . The minor loops are hysteretic, where the emergence of the Hall peak with positive H depends on whether H_{min} surpasses $-H_{\text{peak}}$. The two-component origin of the Hall anomaly peaks can be quite well explained by a distribution of domains having T -dependent H_c using⁴²”

Reviewer #2 (Remarks to the Author):

The author performed systematic study on Cr_2Te_3 thin films on two different substrates, revealing an interesting strain-tunable sign reversal effect of AHE, together with topological Hall effect. Overall, the epitaxy and the characterization are of high quality, especially for the ultra-thin sample (6 unit cells) of strain, a high contrast RHEED pattern was observed clearly. However, I didn't observe the RHEED coming from 3 unit cell sample, as well as the TEM characterization for the sample of 3 unit cell. Since the disappearance of sign change are robustly observed in these sample, I suggest the authors provide the structure analysis on these 3 unit cell sample, before the acceptance of this article's major claim.

Response: We thank the referee for the positive assessment of our manuscript. Following the referee's suggestions, we have now included RHEED patterns for 3 u.c. sample in the revised **Fig. S1c-d**, beyond the originally available XRD in Fig. S2. New TEM result for 3 u.c. is included in **Fig. S3b**.

Also included are new neutron and X-ray reflectometry measurements and analysis, which provide structural, chemical and magnetic characterization of 24 u.c. and 6 u.c. film. This non-destructive probe is considerably more advantageous and comprehensive than a local probe like TEM, particularly in simultaneously obtaining structural and magnetic information from the macroscopic lateral size of the sample with a depth resolution of 0.5 nm.

Although it is beyond the scope of the present work (the following results will be reported in a separate article), the PNR results below illustrate that high-quality Cr_2Te_3 can be grown all the way down to a single unit cell, with well-defined structural (NSLD) and magnetic (MSLD, $M = 37 \pm 3$ emu/cc) scattering length density (SLD) profiles attesting to the film and interface quality.

Reviewer #3 (Remarks to the Author):

Report of the manuscript entitled as “Strain-tunable Berry curvature in quasi-two-dimensional Chromium Telluride” by Hang Chi et al.

Transport behaviors in 2 dimensional materials are attracting considerable attention. Among many different material systems, binary compound of composition $Cr_{(1-\delta)}Te$ is particularly interesting for bridging the band topology and magnetism. On this material system, many intriguing behaviors have been observed, including the giant anomalous Hall effect and temperature-dependent sign change of anomalous Hall effect. On this very topic, through synthesizing MBE grown Cr_2Te_3 thin films, authors studied the nontrivial topology and their contribution to the anomalous Hall effect. In particular, authors identified a hump-like Hall feature, together with a sign-reversal of AHE, which were attributed to the occurrence of spin textures with non-trivial topology.

While the data presented in the manuscript is high-quality, the conclusion is, however, not beyond the those published in the past. In fact, the main results in the present manuscript can be found from literatures. Considering the current development and the novelty of their results, I see no reasons that this manuscript could be published in journals such as Nature Communications. Below I outline my main concerns.

Response: We thank the referee for the critical evaluation of our manuscript and appreciation of the high quality of our data. We concur with the referee that quasi-2D magnetism is a topical, fast evolving field and of particular importance to the condensed matter physics and materials science community. We would like to emphasize that the main results of our work presented here go above and beyond the existing literature.

A. The main feature of our work is the observation of the novel strain-dependent sign change in the AHE and the new DFT theoretical origin. A sign change in AHE accompanied by a nonzero magnetization is a very rare occurrence, and our modeling of it as strain-tunable Berry curvature is original.

B. On the other hand, the hump-shaped Hall features are more common and *not* the main thrust of our work as already noted in the original title of the manuscript, but we have successfully described this feature as arising from strain-modulated magnetic domains.

While a sign change of AHE was observed in the past – no clear physical explanation has been given – one of our achievements is just that, definitely linking the sign reversal AHE phenomenology to the strain-tunable Berry curvature physics. Our new DFT result (now in **Fig. 4c**) uncovering strain-induced AHE sign reversal is a significant novel contribution to the field, which we trust will be well received by the readership of Nature Communications.

The hump-shaped Hall signature, while *not* being our main claim in the original manuscript, may have been observed in some doped/heterostructure-based samples [$CrTe_2/ZrTe_2$, Ou *et al.*, *Nat. Commun.* **13**, 2972 (2022); $CrTe_2/Bi_2Te_3$ Zhang *et al.*, *ACS Nano* **15**, 15710-15719 (2021), Cr_2Te_3/Cr_2Se_3 Jeon *et al.*, *ACS Nano* **16**, 8974-8982 (2022); Cr_2Te_3/Bi nanosheets, Zhou *et al.*, *ACS Appl. Mater. Inter.* **12**, 25135-25142 (2020); Cr_2Te_3/TI , Chen *et al.*, *Nano Lett.* **19**, 6144-6151 (2019)] or a completely different phase of CrTe [Zhao *et al.*, *Nano Res.* **11**, 3116-

3121 (2018)]. The observation of this Hall behavior in *pristine* Cr₂Te₃ is brand new and warrants publication.

Furthermore, several researchers may have fallen prey to the practice of treating the hump-shaped Hall feature as a transport smoking gun for identifying magnetic skyrmions, where, as the referee correctly cautioned, the significance of the scenario involving multichannel conduction should not be overlooked. Thanks to the referee's remark, we now explain this feature as arising from strain-modulated magnetic domains. We have removed all discussion of DMI and topological spin textures as described in more detail in response to Q3 below.

Moreover, we also present new polarized neutron reflectometry (PNR) data that provide the crucial details of the depth profile for the thickness dependent evolution of the magnetic properties. We have performed additional PNR measurements at various magnetic fields, which are now included in the revised **Fig. 2c**. The data show that with reducing magnetic field the depth dependent magnetization M is not uniform and shows a lower M close to the substrate where the strain is higher, and a higher M further from the substrate. Given that PNR is sensitive to the in-plane magnetization, these results directly confirm the scenario of two component distribution. In this case the region with the higher OOP anisotropy will show a lower MSLD measured with an in-plane magnetic field. In addition, we have included PNR data for a 6 u.c. sample, measured with 1 T at 5 K and 60 K, which further corroborates a lower in-plane M compared to 24 u.c. sample due to the stronger OOP anisotropy.

We are therefore confident that our consideration of the referee's specific comments below will lead to a positive decision on the publication of our work by this referee, too.

1: High-quality Cr_(1-δ)Te films have been synthesized by MBE. From this aspect, the material system is lacking of sufficient novelty.

Response: We thank the referee's comments while noting that we did not claim "novelty" on the synthesis itself. However, it is worth emphasizing, that a deep and thorough control and characterization of film growth and understanding of the basic materials properties, obtained through our comprehensive structural, magnetic and transport characterization, are key prerequisites in advancing physics, particularly involving a real magnetic system as complex as Cr₂Te₃.

2: Bump and dip features associated with the intriguing anomalous Hall effect in CrTe films have also been frequently reported. Their possible connection with both the real-space and reciprocal space topology has been suggested and discussed. Thus, this manuscript did not advance the current understanding on the intriguing anomalous Hall effect in CrTe films.

Response: Existing literature pertaining to the bump/dip feature in the broad Cr-Te family of materials is limited to either different phases (CrTe) or modified/heterostructures (Cr₂Te₃/Bi, Cr₂Te₃/Bi₂Te₃, CrTe₂/Bi₂Te₃, CrTe₂/ZrTe₂). In our comprehensively characterized, phase confirmed, and *pristine* Cr₂Te₃, we present brand new data and make an essential contribution to advance the understanding of the AHE phenomenology in chromium telluride in general. The primary message of this manuscript is *not* about the bumps and dips in the AHE curves, but rather about the AHE sign reversal and its explanation in terms of the strain modified bands and Berry curvature. This is *new*.

3: *The sign change of anomalous Hall effect in CrTe films has also been previous reported.*

While the experimental part of the current manuscript is probably high-quality and systematic, the interpretation is lack of sufficient experimental evidence, especially the identification of nontrivial real space topological spin textures and band topology. In this sense, I don't believe it meets the publishing criteria of Nature Comm.

Response: We thank the referee's critique. However, the key claim of our manuscript is not "the identification of nontrivial real space topological spin textures". As emphasized in the original title, it is the non-trivial Berry curvature in the momentum-space that leads to the sign reversal in AHE when modulated by strain. To avoid misreading, we have now modified the manuscript accordingly, particularly by adding novel DFT results that corroborate this strain-tunability of the AHE. Furthermore, we have removed all mentioning of real space topological spin textures since they are not needed to explain the data.

A: This manuscript needs a careful proof reading.

Response: We thank the referee for the suggestion and this has been carefully done.

B: Authors mentioned that "sizable spin-orbit interaction" and "Dzyaloshinskii-Moriya interaction" can be found in the current material system. However, no estimation of both parameters is attempted.

Response: The DFT calculations provide the projected spin-orbit coupling matrix elements. The magnitudes of the *d*-orbital matrix elements are negligible compared to those of the *p*-orbital matrix elements. Parameterizing the non-zero *p*-orbital matrix elements in the usual way such that $|\langle p_i | H_{so} | p_j \rangle| = \frac{\Delta_{so}}{3}$, where Δ_{so} is the energy splitting between the $J = 3/2$ and $J = 1/2$ states, we obtain $\Delta_{so} = 103$ meV for the Te atoms and $\Delta_{so} = 10$ meV for the Cr atoms.

We did not attempt to calculate the strength of the DMI, but we would expect it to be relatively small. The randomness of the sites for the interlayer Cr atoms breaks global inversion symmetry. However, the local bond network that determines the DMI preserves inversion symmetry, as shown in Figure R1 for an interlayer Cr1 atom. This also holds true for the Cr atoms within the CrTe₂ layers. Thus, we would expect the bulk DMI to be small. Surface effects always exist, however there are several things to consider. The energetically most favorable surface is a CrTe₂ tri-layer terminated with an outer Te layer. Thus, within the local bonding network of the outer tri-layer, inversion symmetry is still preserved. Specifically, each Cr is connected to a nearest neighbor Cr via two Cr-Te-Cr bonds that are 180° opposite each other. Thus, the two contributions from the two Te atoms to the DMI between the two Cr atoms will cancel. Finally, the interfaces are not in proximity with heavy metals such as W with large SOC, but with either SrTiO₃ or Al₂O₃, which have relatively small SOC compared to the heavy metals used to induce large interfacial DMI. All of the above considerations lead us to conclude that the DMI should be relatively small in these films.

Therefore, in response to the question about the DMI, we have removed all discussion about DMI and topological spin textures in the manuscript. That discussion was unnecessary, since the presence of simple magnetic domains is sufficient to explain the features in the data as discussed in the text following Fig. 5.

C: While I appreciate the detailed structural characterization, I am not quite sure if they serve for the purpose of “strain-tunable”. In fact, similar behaviors can be observed on both substrates, and authors failed to tune their observation by any other means.

Response: The strain-induced tunability in the AHE refers to the sign change as clearly demonstrated in the new DFT results in **Fig. 4c**. The experimental significance is now better highlighted in new **Fig. 3d**. Both STO and sapphire substrates lead to the same sign of compressive strain, hence attests to the DFT prediction and universality of our observation.

D: Depending on the stoichiometry, Cr₂(1-δ)Te compounds contain rich phases. While authors claimed the composition of their thin films is Cr₂Te₃, the two-step magnetometry data suggest their thin films is not single phase. In fact, shapes of hysteresis loops are lacking of any meaning discussions. Competing contributions from different structural phases could substantially complicate the data analyses. The discussion of blocking temperature needs to be re-examed, which may well suggest the structural inhomogeneity in the system. Thus, from the experimental data, I am not convinced their bump and dip features contain a topological origin.

Response: We thank the referee’s insightful observation. First, we are not suggesting a topological origin of the bump and dip features. Second, in order to investigate the magnetic properties, we performed additional PNR experiments at various values of magnetic fields. Being a depth-sensitive technique, it provides detailed information about the evolution of the magnetization profile as a function of the magnetic field. As we already answered to referee #1, additional discussion pertaining to the thickness dependent evolution of the magnetic properties is now included, suggesting two component contribution – one from the layer farthest from the substrate (lowest strain) showing the a higher H_c ; the other one from the sublayer of the film close to the substrate (higher strain), leading to the lower H_c .

The new data in **Fig. 2c** reveal that with reducing magnetic field the magnetization M is not uniform and shows a lower M close to the substrate where the strain is higher and a higher M further from the substrate. Given that PNR is sensitive to the in-plane magnetization, these results directly confirm the scenario of two component distribution. In this case the region with a higher OOP anisotropy will show a lower MSLD measured with in-plane magnetic field. In addition, we included PNR data for 6 u.c. sample, measured with 1 T at 5 K and 60 K, which consistently show a lower in-plane M compared to 24 u.c. sample due to a stronger OOP anisotropy.

E: Neither an experimental reveal of nontrivial band topology by using ARPES, nor an experimental identification of real-space spin topology are being made. From an experimental perspective, very limited new messages are delivered from this manuscript.

Response: We thank the referee’s suggestions. In the momentum-space, we have obtained new DFT results corroborating the strain-driven effect. Since the material is a metal, and the

interesting topological bands are at the zone edges, it is not clear how ARPES could provide information relevant to band topology. Claims of real space spin topology have been removed.

F: How to estimate the spin polarization of CrTe film? Did they attempt to experimentally measure the possible existence of topological spin textures, since the size of which could be affected by temperature.

Response: We thank the referee's question. The spin polarization parameter was assumed to be 100% for the order-of-magnitude estimation using Eq. 5. However, Eq. 5 and claims of real space spin topology have been removed.

In summary, on a rather well-studied material system, authors revisited the intriguing anomalous Hall effect in CrTe films, and attributed their observations to the combined contribution from band structure topology and real-space spin topology. Their conclusions are however, lacking of sufficient experimental evidences. I am against its publication.

Response: We thank the referee for evaluating our manuscript and trust that we have addressed in detail all the questions of the referee and provided exhaustive explanations, and that the revised version deserves publication.

Reviewers' Comments:

Reviewer #1:

Remarks to the Author:

The authors have addressed my concerns. The novelty of this study is now clearly focused to reveal strain-related sign reversal AHE on Cr₂Te₃.

I recommend publication.

Reviewer #2:

Remarks to the Author:

The data provided by the authors answered my concerns on the 3 nm sample. The sample is of a single crystalline structure and a sharp interface with substrate. Therefore, currently I accept the major claim of this article, and believe that this article may spur more research for the strain tunable band topology for this system.

Reviewer #3:

Remarks to the Author:

Second report of the manuscript entitled as "Strain-tunable Berry curvature in quasi-two-dimensional chromium telluride" by Hang Chi.

I appreciate authors take substantial efforts in addressing some of my comments and comments from the other two reviewers. I could also see a large amount of high-quality data in this revised manuscript. However, as I mentioned in my first report, the anomalous Hall effect studies in the present material system, CrTe and related compounds are abundant. Before making any conclusion, we should discuss how this manuscript could clarify some of the misunderstandings in the past, and advance the physics of CrTe family in the future.

1: Author performed substantial structural characterizations, and based on which authors concluded that "the overall chemical composition of the thin film is uniform". In addition, authors also performed polarized neutron diffraction, which revealed a nonuniform magnetization profile. Let's agree upon this fact.

2: Authors, however, did not explain the step-like hysteresis loops in the main text. Such hysteresis loop is typical for systems with strong inhomogeneity and multiphases. I am pretty sure some of the coauthors understood this phenomenon extremely well. I would ask authors to state explicitly in the main text what causes the step-like magnetic hysteresis loop. In other words, the strain-induced magnetic properties are not being touched.

3: A close connection between the strain-induced magnetic inhomogeneity, and the so-called Berry curvature is not being discussed. The band structure calculation cannot serve as direct evidence. I am wondering if one can possibly calculate the strain-induced magnetic properties and compare it with the experimentally measured magnetic hysteresis loop. I assume strain should be treated as a thickness dependent parameter, since which relaxes from the substrate. This could be more convincing to this referee. Upon this verification, this referee may be in the position of supporting the publication.

4: On a separate note, I have also noted this is a 2D material, the interlayer coupling could be very weak. This suggests the effect of interfacial strain could be limited to a rather narrow range. I am thus curious to see such drastic change in the magnetic hysteresis loop. Can authors comment on this aspect?

5: If the magnetic properties of the bottom layers are different from the rest of the film, one can naturally think its transport properties are also different. This could explain the superposition of two different components in the AHE measurement. This part requires a revisit.

Color/font key - *Questions* are copied in italic, with response in blue, text of previous version in black and new revision in red.

REVIEWER COMMENTS

Reviewer #1 (Remarks to the Author):

The authors have addressed my concerns. The novelty of this study is now clearly focused to reveal strain-related sign reversal AHE on Cr₂Te₃.

I recommend publication.

Response: We thank the referee for recognizing the very clear novelty of the strain-related sign reversal AHE in Cr₂Te₃ and recommending for immediate publication.

Reviewer #2 (Remarks to the Author):

The data provided by the authors answered my concerns on the 3 nm sample. The sample is of a single crystalline structure and a sharp interface with substrate. Therefore, currently I accept the major claim of this article, and believe that this article may spur more research for the strain tunable band topology for this system.

Response: We thank the referee for the well-deserved acceptance of our work on strain tunable Berry curvature, and the strong endorsement for its impact towards spurring future research.

Reviewer #3 (Remarks to the Author):

Second report of the manuscript entitled as “Strain-tunable Berry curvature in quasi-two-dimensional chromium telluride” by Hang Chi.

I appreciate authors take substantial efforts in addressing some of my comments and comments from the other two reviewers. I could also see a large amount of high-quality data in this revised manuscript. However, as I mentioned in my first report, the anomalous Hall effect studies in the present material system, CrTe and related compounds are abundant. Before making any conclusion, we should discuss how this manuscript could clarify some of the misunderstandings in the past, and advance the physics of CrTe family in the future.

Response: We thank the referee’s appreciation of the high quality of our results and the positive consideration that our work is beneficial to the community, as it timely serves to clarify “*misunderstandings in the past*”, [in particular, (i) provide a unified and solid theoretical picture of strain-driven Berry curvature for mechanism of the AHE sign change and (ii) illustrate the importance of two-component scenario underlying the hump-shaped Hall feature, unlikely due to skyrmion-induced THE]; while identifying impactful prospects in the physics-rich Cr-Te family.

1: Author performed substantial structural characterizations, and based on which authors concluded that “the overall chemical composition of the thin film is uniform”. In addition, authors also performed polarized neutron diffraction, which revealed a nonuniform magnetization profile. Let’s agree upon this fact.

Response: We thank the referee’s recognition of our thorough structural characterizations, in particular the depth-sensitive polarized neutron *reflectometry* (PNR) technique we have employed to construct a solid foundation to understand the intricate physics in this rather complex magnetic system. PNR has offered unique insights and irrefutable evidence on the depth dependent evolution of the magnetization profile, corroborating the significant effect of interface-driven strain.

2: Authors, however, did not explain the step-like hysteresis loops in the main text. Such hysteresis loop is typical for systems with strong inhomogeneity and multiphases. I am pretty sure some of the coauthors understood this phenomenon extremely well. I would ask authors to state explicitly in the main text what causes the step-like magnetic hysteresis loop. In other words, the strain-induced magnetic properties are not being touched.

Response: We thank the referee’s suggestion and now provide additional clarifying statement on the multistep switching behavior, further expanding our *existing* concise discussions regarding these issues from the previous review, for example:

- a. Page 4, first revision, on hysteresis loop: “The low- T zero-field kink in the OOP $M(H)$..., the presence of interfacial strain-induced multiple magnetic domains ...”
- b. Page 5, first revision, on strain-induced magnetism: “... a non-uniform depth-dependent magnetization profile with two components, revealing ... more pronounced strain at the interface leads to a higher OOP magnetic anisotropy ...”.

It now reads on Page 4: “The low- T zero-field kink **feature** ³¹⁻³³ in the OOP $M(H)$ becomes more prominent with reduced thickness (see Supplementary Fig. 4 [a new addition in the second revision] for two-component analysis, as well as Supplementary Fig. 5 for additional data on $t = 6$ u.c.). **The multistep hysteresis attests to the presence of varied layer dependent magnetic anisotropies, despite the overall chemical and phase homogeneity of the films** ³⁴. This is consistent with the interfacial strain driven magnetic profiles revealed by the depth-sensitive polarized neutron reflectometry (PNR, **Fig. 2c**) as described below.”

3: A close connection between the strain-induced magnetic inhomogeneity, and the so-called Berry curvature is not being discussed. The band structure calculation cannot serve as direct evidence. I am wondering if one can possibly calculate the strain-induced magnetic properties and compare it with the experimentally measured magnetic hysteresis loop. I assume strain should be treated as a thickness dependent parameter, since which relaxes from the substrate. This could be more convincing to this referee. Upon this verification, this referee may be in the position of supporting the publication.

Response: We thank the referee’s suggestion and note however that the connection among strain, Berry curvature (a well-established and powerful theoretical tool for probing topological, magnetic and AHE physics in novel quantum materials), magnetism and AHE transport is self-evident, unambiguously supported by the comprehensive synergy between DFT-based simulations and structural, magnetic (VSM + PNR) and transport characterizations. We note that Berry curvature is highly sensitive to subtle changes in the electronic band structure, and therefore can be finely tuned by external stimulations, in particular epitaxial strain [e.g., as demonstrated in SrRuO₃ thin films, *PNAS* **118**, e2101946118 (2021)].

We have now elaborated in this regard more explicitly on Page 6, “... This is a highly intriguing transport behavior present in various chromium telluride family of materials and related heterostructures^{31,38-45}, hitherto without consensus on a theoretical origin, yet strikingly similar to the anomaly in SrRuO₃ with a nontrivial band topology²³ and tunability of the Berry curvature via e.g. epitaxial strain^{46,47}. The sensitive interfacial strain dependence of the unique sign reversal behavior of the AHE in Cr₂Te₃ is illustrated in Fig. 3d. ...”

The referee is exactly right the strain is maximized at the film/substrate interface (bottom) and relaxes further into the film (top). Based on the magnetic depth profile revealed by the PNR experiments (Fig. 2c), it is completely justified to treat the strain effect with a two-component model, which is simple yet physically intuitive and sufficient. The measured magnetic hysteresis loops can be readily reproduced by such two-component model. This is now added as new Supplementary Fig. 4.

Supplementary Figure 4 | Interface-driven two component magnetic switching in Cr₂Te₃. a, Field dependence of normalized magnetization M/M_s at 2 K, under the out-of-plane (OOP) configuration for $t = 24, 6$ and 3 u.c. (black, green and orange open symbols, vertically shifted for clarity), respectively. The two-component numerical model (solid line) depicts well the contributions from α (dashed line) and β (dotted line) layers with different magnetic anisotropies modulated by interfacial strain, via $M/M_s \sim \gamma \tanh[\alpha(H + H_{c,\alpha})] + (1 - \gamma) \tanh[\beta(H + H_{c,\beta})]$. b-d, Thickness dependence of γ , the relative weight of the magnetically harder top α layer, as well as the coercive fields $H_{c,\alpha}$ (c) and $H_{c,\beta}$ (d), reveals the more dominant role of the more strongly strained bottom β layer at reduced thickness t .

4: On a separate note, I have also noted this is a 2D materials, the interlayer coupling could be very weak. This suggests the effect of interfacial strain could be limited to a rather narrow range.

I am thus curious to see such drastic change in the magnetic hysteresis loop. Can authors comment on this aspect?

Response: We thank the referee's thoughtful question. As already illustrated in Fig. 1a, the crystalline structure of Cr_2Te_3 is 3D in nature. The largely ionic bonding originated from the partially filled sparse Cr1 layers is thus enough to sustain the effect of interfacial strain in the range of thickness investigated here (up to 24 u.c.). We have now explicitly noted this on Page 2: "... Bulk Cr_2Te_3 crystallizes in a three-dimensional (3D) lattice with space group $P\bar{3}1c$ (D_{3d}^2 , No. 163), as shown in **Figs. 1a-c**, ..."

Furthermore, we have now also provided more discussion concerning the strain-induced magnetic response in revised Fig. 2 with additional schematics on interfacial-strain modulated spin configuration.

On Page 5: "Remarkably, at reduced IP field $\mu_0H = 0.8$ T and 0.05 T, M develops a non-uniform depth-dependent profile with two distinct regions, possessing a lower (higher) IP magnetization value close to (away from) the substrate. Given that the NSLD depth profile of the Cr_2Te_3 layer is uniform and no changes are detected in the structure and chemical composition of the film, we attribute the reduced IP magnetization approaching the substrate to a canting of the magnetization vector towards the OOP direction (schematically drawn as red arrows in **Fig. 2c**). Since the OOP component of the magnetization vector is parallel to the momentum transfer Q , it is not responsive in PNR³⁷. This is consistent with the observed PMA in the VSM measurements (**Fig. 2b**). These results collectively suggest that the more pronounced strain at the film/substrate interface leads to a higher OOP magnetic anisotropy and hence a lower measured IP MSLD.

The observed depth-dependent magnetization configuration is a result of the competition between the anisotropy energy and the Zeeman energy. Thus, under the IP configuration in the PNR experiments in **Fig. 2c**, with reduced IP external field, the Zeeman energy becomes insufficient to compete with the interfacial-strain-enhanced magnetic anisotropy term, giving in to a quick restoration of a more OOP oriented magnetization vector in the bottom layer. This magnetically soft layer is also responsible for the near-zero field kink in OOP $M(H)$ in **Fig. 2b**, where only a small OOP external field is needed for magnetic switching. To completely flip the magnetically harder top layer in the OOP configuration though, a much higher coercive field is required (Supplementary Fig. 4). This is indeed consistent with the observation of a larger IP magnetization preserved in the top layer under reduced IP external field in **Fig. 2c**."

5: If the magnetic properties of the bottom layers are different from the rest of the film, one can naturally think its transport properties are also different. This could explain the superposition of two different components in the AHE measurement. This part require a revisit.

Response: We thank the referee's insightful observation and suggestion. We have already emphasized and explained the physical picture of the superposition in Fig. 5a during the previous review. Indeed, it originates from the differently strained layers – i.e. bottom vs. the rest (Fig. 2c). It can be captured well using a two-component model supported by the PNR results.

We have now more explicitly discussed this in the main text:

On Page 7, “The interface-induced two component magnetic configuration in thicker films (as revealed by PNR in **Fig. 2c**), unambiguously traces the origin of the AHE results, that the transport in the more strongly strained bottom layer possesses the opposite sign from the rest of the layers, whose competition leads to an anomaly of the AHE sign at finite T . Thus collectively, these results demonstrate that epitaxial strain is the key reason for the sign change of AHE in Cr_2Te_3 films (**Fig. 3d**).”.

On Page 8: “The two-component origin of the Hall anomaly peaks in transport corroborates with the magnetic structure revealed in **Fig. 2c** and Supplementary Fig. 4, which is well explained by the distribution of magnetic multilayers/domains modulated interfacial strain with T -dependent H_c , using ⁴⁹ ...”

In summary, we appreciate the referee’s comments, which helped us further improve the readability of the manuscript, by elaborating more on several aspects of the strain-driven Berry curvature and magnetism for an even broader audience. We trust all comments have been fully addressed and the manuscript is now suitable for publication.

Reviewers' Comments:

Reviewer #3:

Remarks to the Author:

Authors answered all my questions. I am not against its publication.

REVIEWERS' COMMENTS

Reviewer #3 (Remarks to the Author):

Authors answered all my questions. I am not against its publication.

Response: No questions.

We appreciate all referees' comments during the review process.